# LLM-ASSISTED SEMANTIC REASONING FOR OPEN-SET ACTIVE LEARNING

## ABSTRACT

Active learning (AL) methods primarily concentrate on closed-set annotations where irrelevant data is absent. However, real-world applications inevitably contain various forms of irrelevant data. This open-set annotation challenge has been explored in some studies, yet two key issues remain. The first is balancing between selecting maximally relevant data and querying uncertain samples, which often increases the proportion of irrelevant data. The second is the inability to distinguish between relevant and irrelevant samples before any labeling, commonly referred to as the cold-start problem. We tackle these challenges with our method named LaSeR (LLM-assisted Semantic Reasoning), which leverages LLM-generated image descriptions and VLM-based similarity scores to, introduce a metric capable of separating relevant from irrelevant data before labeling, and incorporates diversity in the selected samples to enhance model performance. Subsequently in later AL rounds, as more labeled data becomes available, we transfer this knowledge into a detector model to further improve the efficiency of our selection process. Extensive experimental results demonstrate that our method outperforms state-of-the-art AL approaches, as well as recent methods specifically designed for open-set active annotation on standard benchmark datasets.

## 1 INTRODUCTION

Deep learning has achieved remarkable performance in a large number of complex computer vision tasks (LeCun et al., 2015; He et al., 2016; Kirillov et al., 2023), largely fueled by massive datasets with human-annotated labels. However, producing high-quality annotations at scale is costly and time-consuming (VS et al., 2023; Ayub & Fendley, 2022). Active learning (AL) (Settles, 2009) is a widely used approach to reduce annotation costs by selecting the most informative samples for labeling. Traditional AL methods, however, work well in closed-set settings, where unlabeled data contains known classes only. This assumption, however, is violated in real-world settings, where unlabeled data contains novel classes (Ning et al., 2022). For example, consider a domestic robot tasked with helping set up a table for breakfast in a home environment consisting of a variety of objects. A large number of these objects, such as a toothbrush, a piano, etc., are irrelevant to the task. Traditional AL uncertainty and diversity-based techniques usually tag irrelevant novel class instances as the most informative; thus, querying users to learn about irrelevant objects (Ning et al., 2022). This wastes labeling budget, increases human teaching load, and reduces model performance on relevant objects.

Recent works in open-set AL (OSAL) (Park et al., 2022; Ning et al., 2022; Mao et al., 2024; Zong et al., 2024; Zong & Huang, 2025) have attempted to address the problem of identifying relevant classes in the absence of fully labeled datasets. These methods differentiate between known and unknown categories, and focus on selected samples from the relevant classes. However, unlike traditional AL, these methods rely on a portion of initially labeled data of all relevant object classes in the first round to deal with the cold start problem. This, however, goes against the spirit of AL, where data is unlabeled, and a small number of informative samples must be selected by the model. Additionally, these open-set AL methods continually rely on data selection from irrelevant classes in subsequent AL rounds to effectively sample relevant class instances, leading to inefficient data sampling.

In order to tackle these challenges, we propose a large-language model (LLM) based method to semantically reason on textual information for effective selection of in-distribution (ID), task-relevant samples in each AL round. Unlike prior AL and open-set AL methods, our approach, called **LLM-a**ssisted **Se**mantic **R**easoning (**LaSeR**), does not assume any labeled data initially, and only assume the availability of textual labels of the relevant classes. We even consider a stronger constraint where the model has textual labels for only a partial set of relevant classes. We utilize the semantic reasoning ability of LLMs to generate closely related task-irrelevant classes to improve the model's ability to filter out data from confusing, irrelevant classes. We further generate multiple textual descriptions of class labels for diversity sampling of data instances belonging to relevant classes. We then utilize a vision-language model (VLM) to generate text features for the data generated by the LLM, which are compared with unlabeled images to generate relevance and informativeness scores. In the later AL rounds, we train a CNN-based detector model on labeled relevant and irrelevant data from previous AL rounds, treating labeled irrelevant class images as negative examples. We combine the detector scores with VLM-based scores, which helps improve the identification of known-class samples from the unlabeled open-set in later rounds.Extensive experiments on standard open-set AL datasets demonstrate that our method outperforms existing state-of-the-art (SOTA) methods without utilizing any labeled data in the initial AL round. Additional experiments on a robotics object dataset further demonstrate the effectiveness of LaSeR for real-world applications with limited data and annotations. The paper contributes as follows:

- We propose LaSeR, an LLM-based reasoning framework to tackle the cold-start problem in open-set AL, improve filtering of irrelevant class data, and selection of informative relevant class data. To the best of our knowledge, we are the first to tackle open-set AL without the availability of labeled data or a full label set of relevant classes in the initial AL round.

- We adaptively integrate LLM and VLM-based scores with a traditional CNN-based detector to continually improve relevant data selection during later AL rounds.

- Our experiments demonstrate that LaSeR effectively utilizes the annotation budget on informative, relevant class data samples, resulting in superior performance compared to SOTA methods on standard open-set AL benchmark datasets.

## 2 RELATED WORK

**Active Learning.** The goal of active learning (AL) is to maximize performance gains by querying the most useful examples from an unlabeled pool, obtaining their labels, and training on them (Settles, 2009). Most of the AL methods fall under two main categories: uncertainty-based (Kirsch et al., 2019) and diversity-based (Sener & Savarese, 2017) sampling. Uncertainty-based strategies select samples that the model is most uncertain about using various measure of uncertainty, such as entropy (Ayub & Fendley, 2022; Luo et al., 2013), soft-max confidence (Wang & Shang, 2014), or information gain (Gal et al., 2017), while diversity-based approaches use a variety of methods, such as coreset selection algorithm (Sener & Savarese, 2017) or clustering (Citovsky et al., 2021), to estimate the underlying data distribution. Hybrid methods, such as BADGE (Ash et al., 2020), combine both, selecting samples that are simultaneously uncertain and diverse. However, standard AL assumes that unlabeled data come from the same label space as the labeled set, so when out-of-distribution (OOD) or unknown-class images are present, the uncertainty/diversity criteria tend to over-query them, wasting annotation budget and degrading downstream accuracy.

**Open-Set Recognition (OSR)** seeks to correctly classify known classes while rejecting unknowns at test time. Methods include calibrated classifier heads such as OpenMax (Bendale & Boult, 2016), which uses Extreme Value Theory to assign "unknown" class probability, generative modeling to synthesize or approximate space of "unknowns", e.g., G-OpenMax (Ge et al., 2017), reconstruction-based detectors like C2AE that threshold class-conditioned reconstruction errors (Oza & Patel, 2019), and prototype/reciprocal-point approaches that explicitly model "otherness" around class regions, e.g., RPL (Chen et al., 2020). While these ideas are related to OSAL, OSR typically has all known/relevant class data fully labeled during training, whereas the OSAL setting considered in this paper does not have any labeled data in the beginning. Additionally, OSR methods have no access to irrelevant data during training, while OSAL encounters irrelevant data during the iterative querying process and utilized the knowledge gained from this data to improve performance in later AL rounds. These differences make OSR solutions insufficient on their own for OSAL.

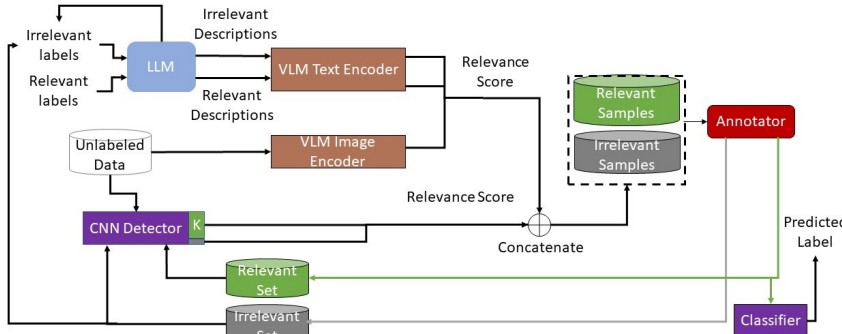

Figure 1: Overview of the LaSeR architecture for OSAL. In each AL round, the LLM and VLM-based scores are concatenated with the detector scores for relevant and informative data selection. After annotation, the detector and the classifier are trained on the complete and relevant labeled data instances, respectively. Irrelevant classes are also added to future LLM prompts.

**Open-Set Active Learning** Recent approaches adapt AL to mixed unlabeled pools containing both in-distribution (relevant) and out-of-distribution (OOD)/irrelevant examples (Ning et al., 2022; Safaei et al., 2024). LfOSA trains a detector alongside the classifier, modeling per-example max-activation values with a GMM and temperature tuning to filter OOD classes and preferentially annotate ID class samples (Ning et al., 2022). MQ-Net (Meta-Query-Net) treats the purity–informativeness trade-off as a meta-learning problem, learning to balance the two as OOD ratios and training stages change (Park et al., 2022). EOAL (entropic open-set active learning) leverages two different entropy scores to effectively select ID and OOD class samples (Safaei et al., 2024). EAOA proposes an energy-based scoring framework that uses both epistemic and aleatoric uncertainty to produce reliable query selection (Zong & Huang, 2025). BUAL combines a normal classifier and a "negative" classifier to bidirectionally score uncertainty so it can query the most informative samples while avoiding unknown-class noise (Zong et al., 2024). All of these OSAL methods still rely on a portion of ID/relevant class dataset to be available in the beginning to deal with the cold-start problem, and they also continually rely on getting annotations for OOD/irrelevant class samples, which wastes annotation budget and negatively affects model performance. In this paper, we propose a novel LLM-based reasoning framework to tackle these challenges in OSAL.

## 3 METHODOLOGY

Open-Set Active Learning (OSAL) extends traditional Active Learning by introducing the challenge of distinguishing between *relevant* and *irrelevant* classes within an unlabeled dataset. Given an unlabeled pool $D_U = \{x_j^U\}_{j=1}^{n^U}$, samples may belong either to the known set of relevant classes $\mathcal{Y}_R$ or to an open set of unknown and potentially irrelevant classes $\mathcal{Y}_I$. Specifically, $D_U = X_{kno} \cup X_{unk}$, where $X_{kno}$ denotes examples from known relevant classes and $X_{unk}$ represents examples from unknown irrelevant classes. In traditional OSAL settings (Ning et al., 2022), an initial small labeled dataset $D_L = \{x_i^L, y_i^L\}_{i=1}^{N_L}$ of known samples is also available. However, in this paper, we consider the OSAL setting where $D_L$ is unavailable and only the set of relevant classes $\mathcal{Y}_\mathcal{R}$ is available[1]. At each iteration, a query set $X_{\text{query}} = X_{\text{query}}^{kno} \cup X_{\text{query}}^{unk}$ of batch size $b$ is constructed from $D_U$ and labeled by the oracle. The goal of OSAL is to train a model $f_{\theta_D} : \mathcal{X} \rightarrow \mathcal{Y}_R \cup \mathcal{Y}_I$ that can effectively differentiate between these two subsets while selectively querying the most informative samples from $X_{kno}$. The labeled $X_{kno}$ in AL rounds is used to train a classifier model $f_{\theta_c}(.)$ with parameters $\theta_c$ for an intended classification task of relevant classes. The subsections below describe the main components of our framework to address this OSAL problem.

---

[1]We further consider a case where only a partial set of relevant classes is available in the experiments (Section 4.5).

### 3.1 LLM-BASED TEXTUAL DESCRIPTIONS OF RELEVANT AND IRRELEVANT CLASSES

Unlike prior OSAL methods, we do not assume the availability of a labeled dataset of relevant classes in the beginning, leading to the cold-start problem. Our goal is develop a method that can rely on the text-based label set of relevant classes $\mathcal{Y}_R = \{y_k\}_{k=1}^K$ only to determine relevant class samples that are the most informative from the unlabeled data pool $D_U = \{x_j^U\}_{j=1}^{n_U}$. To address these challenges, we utilize the semantic reasoning capability of LLMs, and prompt them to generate $M$ number of relevant class descriptions (denoted as set $\mathcal{T}_R$) for each of the $K$ relevant classes to ensure a diverse and informative selection of relevant class instances from the unlabeled data pool. For example, for a class label *cat*, the LLM generated descriptions, such as "A photo of a cat walking", "A photo of a cat lying down", "A cat is sleeping", etc. Utilizing the textual descriptions of relevant classes only leads to sub-optimal performance in the selection of relevant class samples from the unlabeled data pool, and requires the use of textual descriptions for irrelevant classes. However, the model does not have access to the open set of all possible irrelevant classes it might encounter. We again utilize LLMs to generate the $N_{conf}$ most closely related classes for each relevant class. For example, for the class *airplane*, the LLM might select *bird* to be a closely related class. The reasoning behind generating closely related classes is that these can help filter out irrelevant samples in the unlabeled data pool that might be confused with belonging to the relevant classes. Similar to relevant classes, we generate $M$ number of textual descriptions for closely related classes (denoted as set $\mathcal{T}_I$). Figure 2 shows the 2D projection of embeddings generated by a VLM for textual descriptions generated by an LLM for a relevant class *airplane*, and for closely-related, LLM-generated irrelevant classes, illustrating both the diversity of LLM descriptions and the proximity of LLM-generated, closely related negatives. Examples of generated irrelevant classes, and textual descriptions, accompanied by LLM prompts, are described in Appendix A.

### 3.2 VLM-BASED RELEVANCE SCORES

We then utilize the text encoder of a VLM (e.g. CLIP (Radford et al., 2021)) to generate embedding $\phi(t), \forall t \in \mathcal{T}_R$ and $\phi(t'), \forall t' \in \mathcal{T}_I$ for the relevant and irrelevant class descriptions generated by the LLM. For relevant sample selection, we generate an embedding $\phi(x)$ for each image $x$ in the unlabeled data pool $D_U$ and find its relevance score as follows:

$$S_{\text{vlm}}(x) = \max_{t \in \mathcal{T}_R} \cos(\phi(x), \phi(t)) - \frac{1}{|\mathcal{T}_I|} \sum_{t' \in \mathcal{T}_I} \cos(\phi(x), \phi(t')) \tag{1}$$

The first term selects the maximum cosine similarity score between $\phi(x)$ and relevant class text embeddings $\phi(t)$, while the second term penalizes similarity to irrelevant classes via the average similarity score between $\phi(x)$ and irrelevant class text embeddings $\phi(t')$. Samples with the highest $S_{vlm}(x)$ values are selected for annotation, resulting in an informative labeled dataset while minimizing annotation costs, ensuring that the majority of labeled samples belong to relevant classes for training a classifier. Note that the query set will still likely contain irrelevant class instances, allowing the model to get labels for irrelevant classes. These labels are used in later AL rounds to refine LLM-based reasoning for generating irrelevant classes and their textual descriptions for better selec-

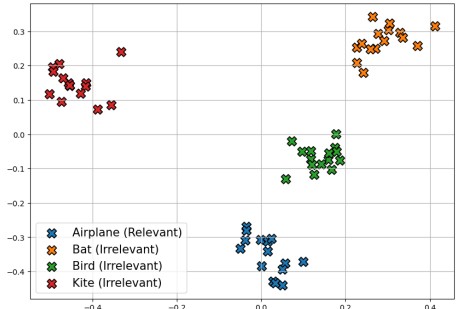

Figure 2: 2D projection of VLM-based embeddings of text descriptions generated by the LLM for a relevant class (Airplane) and closely-related irrelevant classes.

tion of relevant class samples from $D_U$. Also, to ensure that we do not perform redundant sampling from the same relevant textual descriptions, we sample *top-p* relevant samples (with highest $S_{vlm}(x)$ scores) for each textual description $\mathcal{T}_R$, where $p = \frac{b}{M}$, and $b$ is the query batch size in an AL round.

#### 3.2.1 CNN-BASED DETECTOR SCORES

In the first round of OSAL, no labeled examples from irrelevant classes are available. At this stage, LLM and VLM-based relevance score $S_{vlm}(x)$ serves as an effective way to select informative, and

relevant class samples. However, in later AL rounds, two challenges emerge: 1) the distinguishing power of $S_{vlm}(x)$ diminishes and it struggles to separate fine-grained differences between relevant and irrelevant class instances, which reduces selection precision, and 2) after several AL rounds, we accumulate labeled examples from irrelevant classes that provide ground-truth evidence of what constitutes as irrelevant data in the current task, making it inefficient to rely solely on heuristic $S_{vlm}(x)$ scores.

To address these problems, we introduce a CNN-based detector model $f_{\theta_D}$, with parameters $\theta_D$, to be used in combination with LLM and VLM for data selection in later AL rounds. After the first AL round when labeled data from both relevant and irrelevant classes is available, we train $f_{\theta_D}$ that outputs probabilities over $n + 1$ classes, where $n$ corresponds to the relevant classes $\mathcal{Y}_R$ and the additional class accounts for irrelevant classes $\mathcal{Y}_I$. The detector is trained using a cross-entropy loss over the labeled dataset $D_L$ available in that round.

$$\mathcal{L}_{det} = -\frac{1}{|D_L|} \sum_{(x,y) \in D_L} \sum_{c=1}^{n+1} \mathbf{1}[y = c] \log p_\theta(c \mid x). \tag{2}$$

We further employ temperature scaling as in (Ning et al., 2022) to sharpen the probability distributions and improve the separability of known and unknown samples. With the temperature scaling parameter $T$, the predicted probabilities are defined as:

$$q_c^T = \frac{\exp(a_c/T)}{\sum_j \exp(a_j/T)}, \tag{3}$$

where, $a_c$ denotes the activation for class $c$. During the query phase, the unlabeled dataset $D_U$ is passed through the trained detector to get the probability distribution $p(y|x)$ for each unlabeled sample $x$. The relevance score from the detector is calculated by taking the highest probability among the relevant classes and subtracting the probability assigned to the irrelevant class for $x$:

$$S_{\text{det}}(x) = \max_{y \in \mathcal{Y}_r} p(y|x) - p(y = \text{irrelevant}|x). \tag{4}$$

This score favors samples that are confidently assigned to relevant categories while penalizing those that the detector associates with irrelevant data. This score is combined with the relevance score generated by the LLM, VLM stage $S_{vlm}(x)$:

$$S_{\text{final}}(x) = (1 - \delta) \, S_{\text{vlm}}(x) + \delta \, S_{\text{det}}(x), \tag{5}$$

where, $\delta$ (a hyperparameter) controls the contribution of the LLM, LLM module and the detector towards the overall relevance score. In the first round, $\delta = 0$ so that the model relies on the LLM and VLM based scores only. We gradually increase $\delta$ in later AL rounds shifting the balance of the selection process toward the detector. This dynamic weighting ensures that our model exploits the zero-shot semantic reasoning capability of LLMs in the early rounds, and leverage the discriminative power of the detector in later rounds. Pseudocode for LaSeR is detailed in Algorithm 1.

## 4 EXPERIMENTS

For validation of our approach, we use standard OSAL datasets (Ning et al., 2022), such as CIFAR-10 (Krizhevsky, 2009), CIFAR-100 (Krizhevsky, 2009) and Tiny-ImageNet (Yao & Miller, 2015) Datasets, as well as a non-standard robotics dataset, iCubWorld Kirtay et al. (2020) collected using the iCub humanoid robot Metta et al. (2008). CIFAR-10 and CIFAR-100 each contain 50,000 training images and 10,000 test images, covering 10 and 100 classes, respectively. Tiny-ImageNet contains 100,000 training and 20,000 test images across 200 classes. We follow the standard OSAL protocol (Ning et al., 2022) to construct open-set versions of these datasets, by choosing a percentage of the classes as relevant classes and others as irrelevant. We set the mismatch ratio to 20%, 30%, and 40% for all three datasets across all experiments, where this ratio denotes the fraction of known classes among all classes. For example, at 20% on CIFAR-10/100/Tiny-ImageNet, the first 2/20/40 classes are treated as known for training and the remaining 8/80/160 as unknown. Details of the iCubWorld dataset are in Section 4.5.

---

**Algorithm 1** LaSeR for Open-Set Active Learning

---

**Input:** $\mathcal{X}_U$ (unlabeled set), $\mathcal{Y}_R$ (relevant classes), $\mathcal{Y}_I = \Phi$ (irrelevant classes), $N_{conf}$ (# of irrelevant classes), $K$ (# of descriptions per class), $b$ (query batch size), $J$ (# of AL rounds), $\delta = 0$

**Ensure:** Classifier $f_{\theta_c}$ trained on $\mathcal{Y}_R$

1: **for** $j = 1, 2, \ldots, J$ **do**
2:    $\mathcal{Y}_{conf} = \text{LLM}(\mathcal{Y}_R)$, $\mathcal{Y}_{conf} = \mathcal{Y}_{conf} \cup \mathcal{Y}_I$ *#Update irrelevant set with LLM-generated irrelevant classes*
3:    $\mathcal{T}_R = \text{LLM}(\mathcal{Y}_R)$, $\mathcal{T}_I = \text{LLM}(\mathcal{Y}_{conf})$ *#LLM-generated text descriptions*
4:    Get VLM-based embeddings $\phi^{(j)}(t)$, $\phi^{(j)}(t')$, $\phi^{(j)}(x)$, $\forall x \in \mathcal{D}_U$ for text descriptions and images.
5:    Calculate VLM-based relevance score $S_{vlm}^{(j)}(x)$ for $x$ using Eq. (1).
6:    $p(y|x) = f_{\theta_D}(x), \forall x \in \mathcal{D}_U$ *#get detector-based probability distributions for unlabeled data*
7:    Generate detector-based relevance score $S_{\det}^{(j)}(x)$ using Eq. (4).
8:    $S_{\text{final}}^{(j)}(x) = (1 - \delta)S_{vlm}^{(j)}(x) + \delta S_{\det}^{(j)}(x)$ *#Combine VLM and detector scores using Eq. (5)*
9:    $X_{\text{query}}^{(j)} = \arg\max_{\substack{X \subset \mathcal{D}_U \\ |X|=b}} \sum_{x \in X} S_{\text{final}}^{(j)}(x)$ *#Select top-b unlabeled samples*
10:   $Y_{\text{query}}^{(j)} = \text{Oracle}(X_{\text{query}}^{(j)})$ *#Obtain ground-truth labels for top-b samples*
11:   $\mathcal{D}_L^{(j)} = \mathcal{D}_L^{(j-1)} \cup \{(X_{\text{query}}^{(j)}, Y_{\text{query}}^{(j)})\}$ *#Augment labeled pool*
12:   $f_{\theta_c}^{(j)} \leftarrow \arg\min_{f_{\theta_c}} \mathcal{L}_{\text{CE}}(f, \mathcal{D}_L^{(j)}|_{\mathcal{Y}_R})$ *#Train classifier on relevant samples*
13:   $f_{\theta_D}^{(j)}(x) = p(y|x), \quad y \in \mathcal{Y}_R \cup \{\text{irr}\}$ *#Train detector to separate relevant and irrelevant classes*
14:   $\delta \uparrow \quad (j \to J)$ *#Shift reliance from LLM and VLM to detector over AL rounds*
15: **end for**

---

### 4.1 IMPLEMENTATION DETAILS

For all OSAL methods other than ours, random initial sampling of 1%, 8% and 8% on CIFAR-10, CIFAR-100 and Tiny-ImageNet, respectively, is done to deal with the cold start problem, whereas this step is skipped for our method. Across all experiments, we run 10 AL rounds, querying 1,500 samples per round for annotation. For a fair comparison with prior works, such as EOAL Safaei et al. (2024), in each AL round, we train a ResNet18 for 300 epochs via SGD with momentum of 0.9, weight decay of 5e-4, initial learning rate 0.01, and batch size of 128. The learning rate is decayed by 0.5 every 60 epochs. We use the same network and training hyperparameters for both the detector and the classifier, except the detector employs a temperature scaling with $T = 0.5$. In the first AL round, $\delta = 0$, and we increase it by 0.1 in each AL round. We use Pytorch (Paszke et al., 2019) to implement our method and an NVIDIA RTX 4090 GPU for training. We use GPT-4o mini and CLIP (Radford et al., 2021) as the LLM and VLM models, respectively, for LaSeR. $N_{conf}$ (number of LLM-generated irrelevant classes) is set to be double the relevant classes in any experiment. For example, at 30% mismatch ratio on CIFAR-100, $N_{conf} = 60$. The number of generated text descriptions $K$ is set to 15, 4, and 2 for the CIFAR-10, CIFAR-100, and Tiny-ImageNet datasets, respectively. For robustness, we run each experiment four times, varying the split between relevant and irrelevant classes, and report the mean over runs.

### 4.2 BASELINES

We compare our method to other OSAL-focused approaches as well as approaches developed for closed-set AL. We compare *i)* **EAOA**, *ii)* **BUAL** *iii)* **EOAL**, *iv)* **LfOSA**, *v)* **MQNet**, *vi)* **BALD**: it uses uncertainty from Bayesian Inference to select samples. *vii)* **OpenMax**: a representative OSR method. *viii)* **Random**: it selects samples randomly. *ix)* **Uncertainty**: it selects samples with the highest uncertainty. *x)* **LaSer (ours)**: the proposed method. EAOA, BUAL, EOAL, LfOSA, and MQNet are described in Section 2. We report results of all OSAL methods from EAOA and EOAL papers, if available.

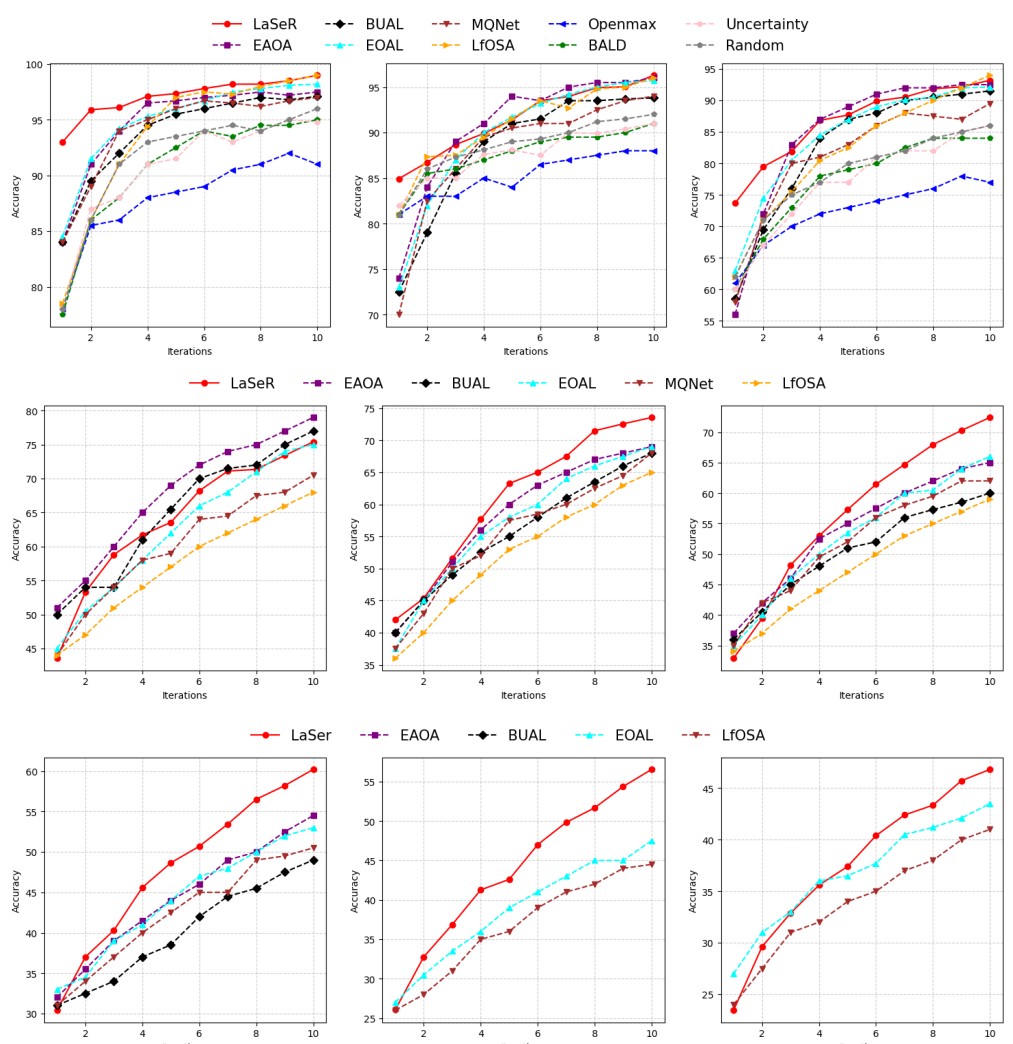

Figure 3: Accuracy results for CIFAR-10 (top), CIFAR-100 (middle), and Tiny-ImageNet (bottom). First, second, and third columns show accuracy plots for 20%, 30%, and 40% mismatch ratios

## 4.3 PERFORMANCE METRICS

We use precision, and accuracy to compare all OSAL methods. Precision is the ratio of relevant samples selected in each AL round to the total number of samples selected to be queried in that round. The classification accuracy is the accuracy achieved by the classifier on the test set for the relevant classes. We also report another metric (recall) in the appendices.

## 4.4 OSAL RESULTS

Figures 3 compares the classification accuracy of LaSeR with SOTA OSAL methods on CIFAR-10, CIFAR-100, and Tiny-ImageNet. For all datasets, the importance of mitigating the cold start problem is evident in earlier AL rounds when less labeled data is available. LaSeR outperforms other methods by a significant margin in the earlier rounds. For example, the performance gap between LaSeR and the next best method in the first round on CIFAR-10 with 20% mismatch ratio is ∼9%. In the later AL rounds, for CIFAR-10 on all mismatch ratios, there is a saturation of labeled samples, which results in LaSeR and other SOTA OSAL methods, EAOA, BUAL, EOAL, MQNET, and LfOSA achieving ∼99% accuracy. For CIFAR-100 and Tiny-ImageNet, however, since samples for each class are lower in number, we do not get saturation of data even in the later rounds. In this

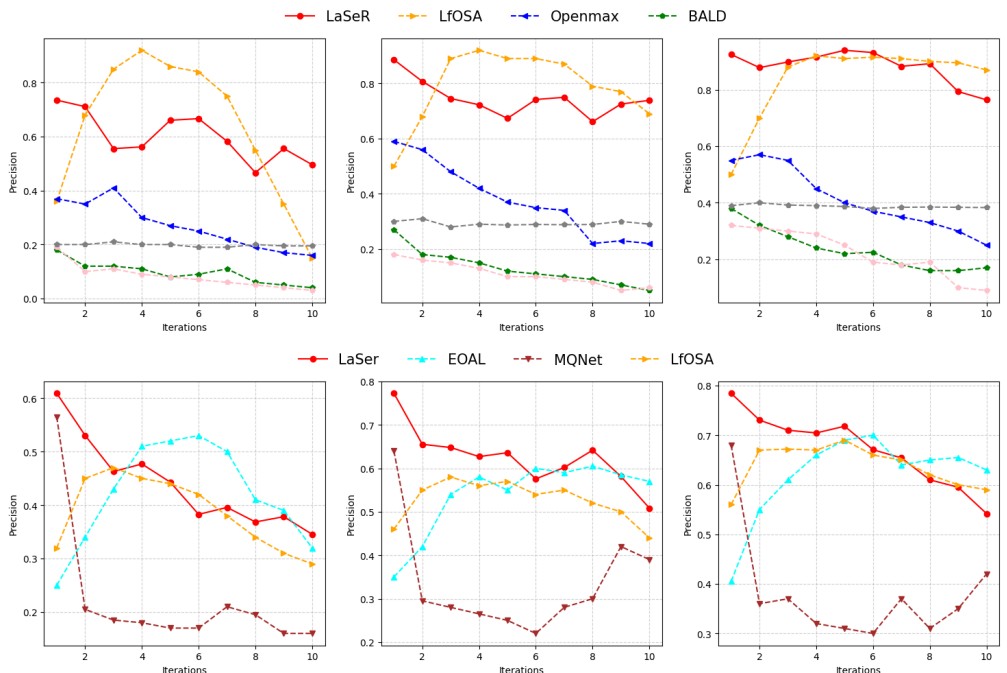

Figure 4: Precision results for CIFAR-10 (top), and CIFAR-100 (bottom). First, second, and third columns show precision plots for 20%, 30%, and 40% mismatch ratios

case, LaSeR consistently outperforms SOTA methods on all AL rounds on all mismatch ratios for both CIFAR-100 and Tiny-ImageNet, except CIFAR-100 with 20% mismatch ratio where EAOA achieves 3% higher accuracy than LaSeR. Particularly, LaSer outperforms the next best method (EAOA and EOAL) by margins of ~5% and ~7% in the last round on CIFAR-100 for 30% and 40% mismatch ratios, respectively. Similarly, LaSeR outperforms the next best method (EAOA and EOAL) in the last AL round on Tiny ImageNet with all mismatch ratios by margins of ~5%, ~10%, and ~3%, respectively.

Figures 4 compares the selection precision of all methods on the CIFAR-10, and CIFAR-100 datasets. Similar to classification accuracy results, LaSeR achieves significantly higher precision compared to all other methods in the initial AL rounds for all settings, as the LLM, and VLM-based relevance scores help select the most relevant class instances. For CIFAR-10, LfOSA achieves higher precision in middle AL rounds, as it mainly focuses on selecting relevant samples only without considering the informativeness of the samples, leading to high precision but at the cost of sampling uninformative samples. In contrast, LaSeR's precision starts to drop as the advantage of LLM and VLM-based scores to select relevant samples diminishes. However, by the end of all AL rounds, LaSeR outperforms LfOSA in most settings, as the CNN-detector starts to select relevant samples by relying in previously annotated data. For CIFAR-100, similar pattern is observed for LaSeR, but instead of LfOSA, EOAL is the next best method exhibiting a similar pattern to LfOSA on CIFAR-10. Overall, LaSeR achieves high precision in the initial rounds to select the most informative and relevant class samples, and then strikes a balance between selecting informative, relevant, and irrelevant samples to help improve detector performance, and maintain a relatively high precision across all AL rounds. These results further demonstrate the effectiveness of our method across multiple datasets and mismatch ratios, particularly in the initial AL rounds. Precision results were not reported by any other method on Tiny-ImageNet, and thus not included here.

## 4.5 iCubWorld Results

The introduction of VLM and LLM also introduces significant prior knowledge in terms of pre-trained large models. This requires validation of our approach on non-standard datasets. To demonstrate the generalizability and real-world applicability of LaSer, we conducted an experiment on

| Iteration | 1 | 2 | 3 | 4 | 5 | 6 | 7 | 8 | 9 | 10 |
|---|---|---|---|---|---|---|---|---|---|---|
| Accuracy (%) | 62.7 | 76.4 | 81.5 | 93.6 | 94.5 | 95.8 | 96.6 | 97.3 | 97.1 | 97.3 |
| Precision | 0.88 | 0.71 | 0.52 | 0.58 | 0.62 | 0.63 | 0.49 | 0.59 | 0.65 | 0.57 |

Table 1: LaSeR results on the iCubWorld dataset with 20% mismatch and partial relevant class labels.

the iCubWorld dataset (Kirtay et al., 2020). The dataset was collected using the iCub humanoid robot Metta et al. (2008) and it includes 20 major categories and 10 items in each category. Each item consists of $\sim$3600 256$\times$256 images. Given the redundancy in the images per object, we select a random mixture of images per object, resulting in a total of 65000 images equally divided among 200 object items.

To demonstrate the practicality of our approach, we consider that the robot is deployed in a home environment and and it must perform the task: "Help the person in dressing up". We then select two classes that are relevant to this task:"Perfume","Sunglasses" as a partial set of relevant classes known to the model. We then simulate the case where the robot can explore it's unlabelled environment, and prompt the LLM to detect more relevant classes, which results in 2 more LLM-generated relevant classes. We generated 2 relevant classes only for practicality reasons to make this a 20% mismatch ratio experiment. "Hairclip" and "Hairbrush" among the available set of classes. We then evaluate LaSeR on the OSAL experiment for 10 AL rounds, similar to how it was evaluated on standard datasets. Table 1 shows that LaSeR reaches 90-95% accuracy within the first 4 AL rounds and then gets saturated. The selection precision also starts higher at $\sim$88% and then reaches $\sim$57% in later rounds as the dataset saturates. These results demonstrate the practical usage of LaSeR for robotics applications where data and annotation budget might be limited, and a complete knowledge of the relevant and irrelevant classes might be unknown.

## 4.6 ABLATION STUDIES

To analyze the contribution of our approach, we conducted an ablation study on CIFAR-100 dataset with a 20% mismatch ratio with 50 epochs. We consider the following variations of our method for the ablation study:

- **Without Detector** indicates that the detector model was not used in our method and it relied on $S_{\text{vlm}}(x)$ scores only.
- **N = 0** indicates that the no irrelevant classes were generated by the LLM when calculating $S_{\text{vlm}}(x)$.
- **K = 1** indicates that only 1 description was generated by the LLM for each relevant and irrelevant class when calculating $S_{\text{vlm}}(x)$.
- **K = 0, N = 0** indicates that LLM part was completely discarded. No irrelevant class names and textual descriptions were generated. Only the relevant class names were used when calculating $S_{\text{vlm}}(x)$ using the VLM.
- **K = 0, N = 0, Without detector** indicates that detector part was also removed on top of removing the LLM part, and it relied on $S_{\text{vlm}}(x)$ scores only.
- **Precision Based Delta** indicates that $\delta$ was not incremented by 0.1 in our experiments. For each iteration, it was calculated using the formula: precision of detector / (precision of detector + precision of VLM). Here precision refers to the selection precision of the LLM-VLM and detector components individually in the previous AL round.

Figure 5 shows the classification accuracy and precision for different ablations of our method. Precision scores show that when removing the detector, $S_{\text{vlm}}(x)$ alone cannot select samples as efficiently after the first AL round. By the last round, precision is $\sim$6% lower and accuracy is $\sim$3% lower compared to the complete LaSeR method.

The removal of LLM-generated irrelevant classes in the calculation of $S_{\text{vlm}}(x)$ results in $\sim$5% lower precision in the first AL round, and the gap stays similar in the last AL round. Similarly, in the first round, the gap between accuracy for LaSer and N = 0 is $\sim$6% and it widens to $\sim$10% in the last

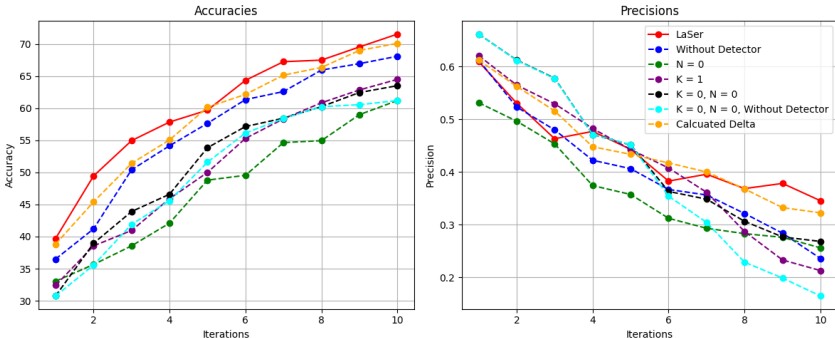

Figure 5: Accuracy (left) and precision (right) results of the ablation Study on CIFAR-100 with 20% mismatch ratio

round. The reason is that, without penalizing irrelevant class similarities, many samples similar to known classes get higher similarities with relevant class descriptions and end up in the query set.

For K = 1, the model shows ∼6% lower accuracy through all AL rounds, even though the precision is comparable to LaSer early on. Lower accuracies for $K = 1$ are due to the lack of diversity in selected samples for annotation, as a higher $K$ value would mean more variant text descriptions, leading to a diverse selection of samples. When $K$ is fixed to 1, all the selected samples are similar to each other and lack any informativeness. This also explains why precision for $K = 1$ is a bit higher in a few initial AL rounds compared to LaSeR. Based on these results, the generation of irrelevant classes is the most important component of the method, followed by the generation of variant text descriptions, and finally the integration of the traditional CNN detector in LaSeR.

The removal of the LLM completely leads to ∼10% decline in accuracy compared to the complete LaSeR model, demonstrating the need for generating diverse textual descriptions and irrelevant classes. When the detector is also removed, and the model relies on a VLM only, the precision is the highest in the earlier AL rounds, as the model selects relevant class samples effectively, but the accuracy is significantly lower than LaSeR, indicating low diversity in selected data as diverse text descriptions are not available.

Figure 6 shows the values of $\delta$ for both variants with varying $\delta$. At the beginning of training, when the detector first starts learning, the precision-based $\delta$ increases rapidly. In later rounds, however, it stabilizes as the detector precision improves more slowly. In this setting, the resulting accuracy and precision are relatively similar across the two cases. Yet, in the early AL rounds, when the precision-based $\delta$ is larger than the increment-based one, we observe that constraining the detector's contribution to a smaller proportion in LaSeR leads to higher accuracy.

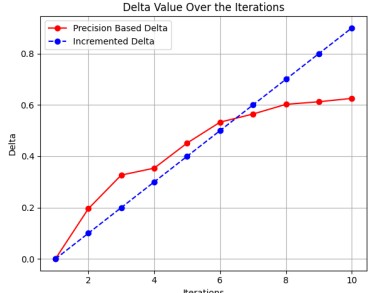

Figure 6: Values for $\delta$ in AL rounds

## 5   CONCLUSION

In this paper, we present a novel framework, termed LaSeR, to address the open-set active learning problem. Unlike prior works, we do not rely on any labeled data initially to mitigate the cold start problem. We utilize the semantic reasoning ability of LLMs and the vision and language alignment ability of VLMs to tackle this problem and further improve the selection of relevant class samples. Additionally, we adaptively integrate the LLM and VLM-based scores with traditional CNNs to effectively utilize annotated data in previous AL rounds to continually improve the selection of relevant and informative class samples in later AL rounds. Experimental results on multiple datasets demonstrate that LaSeR can effectively use the query budget on selecting relevant class samples throughout the AL rounds, which results in significantly higher classification accuracy and precision than the SOTA OSAL methods.

## REPRODUCIBILITY STATEMENT

Algorithm 1 in Section 3 provides the pseudocode of the proposed method, and Section 4 provides all the implementation details of our method, including the datasets, experimental settings, and values chosen for the hyperparameters. Finally, appendix A provides exact prompts used for the LLM to generate irrelevant class labels and text descriptions of classes used in our method. We commit to publicly releasing the complete code base of our method after the paper is accepted.

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

## A LLM PROMPTS

This section contains the full prompts used for irrelevant class names generation and class descriptions generation. The prompt is in grey part followed by a test example and then a highlighted response from the LLM (GPT-4o mini).

## A.1 PROMPT FOR GENERATING IRRELEVANT CLASS NAMES

> **Prompt**
>
> You are given a list of relevant class names for a task. Your goal is to generate {N} new class names that are not in the list but could be easily confused with them in a real-world environment. The generated class names should: Be semantically or visually similar to the relevant class names. Represent plausible but distinct categories. Avoid duplicates or trivial variations (e.g., just adding numbers or "object"). Stay within the same domain or environment as the relevant classes. Input:
> Relevant class names: {class_names}
> Number of class names to generate: {N}
> Output: A list of {N} new class names that are potentially confusing with the given relevant classes.
> Example Input: Relevant class names: ["cat", "dog", "horse"] Number of class names: 5
> Output: Irrelevant classes = ["wolf", "coyote", "tiger", "donkey", "goat"]

**Test Example:** Relevant classes = ["Airplane", "Car"] , N = 5

> **LLM Response**
>
> Irrelevant classes = ["Bird", "Bat", "Truck", "Kite", "Bus"]

## A.2 PROMPT FOR GENERATING CLASS DESCRIPTIONS

> **Prompt**
>
> You are an helpful assistant that can generate sentences for a given object in different poses or doing different things. You will be given a list of objects and your task is to generate generic sentences in which those objects are doing some things or interacting with their environment in some way. Output format: {'object1': ['object1 is standing', 'object1 is sitting', 'object1 is tilting'], 'object2': ['object2 is on table', 'object2 is idle', 'object2 is moving'], 'object3': ['object3 is on being used', 'object3 is doing nothing', 'object3 is making noise']} Example: input: ['boat', 'hand', 'frog'] output: 'boat': ['a boat is in water and it is idle', 'someone is riding a boat and the boat is in water', 'a boat is being ridden by someone and their is blue sky behind it'], 'hand': ['a person is showing a hand to someone', 'a man is holding a hand and something in it', 'a hand with 5 fingers and something behind it'], 'frog': ['a frog is jumping and something is behind it', 'a frog is swimming and the water is blue', 'a frog is idle and standing on something'] Remember: You are giving sentences that can describe very simple images so keep the sentences very simple and in format that may be used to describe images. You can be given more than 3 objects Remember: Example is of 3 sentences for each class but you have to give {K} sentences for each class. Following are the classes: {classes} give me sentences in json form.

**Test Example:** classes = ["Bat", "Bird", "Kite"] , K = 5

> **LLM Response**
>
> Class Descriptions = {"Bat": ['a bat is hanging upside down', 'a bat is flying at night', 'a bat is resting on a branch', 'a bat is catching insects in the air', 'a bat is standing on a tree'], "Bird": ['a bird is perched on a wire', 'a bird is flying in the sky', 'a bird is building a nest', 'a bird is singing on a branch', 'a bird is searching for food on the ground'], "Kite": ['a kite is soaring high in the sky', 'a kite is being flown by a child', 'a kite is dancing in the wind', 'a kite is stuck in a tree', 'a kite is coming down for a landing']}

|  | CIFAR-10 | | | CIFAR-100 | | |
|---|---|---|---|---|---|---|
| Mismatch Ratio | LaSeR | Llava | Gap | LaSeR | Llava | Gap |
| 20% | **99.0** | 95.2 | +3.8 | **75.0** | 52.8 | +22.2 |
| 30% | **97.0** | 94.4 | +2.6 | **73.5** | 43.2 | +30.3 |
| 40% | **93.0** | 91.7 | +1.3 | **72.5** | 40.4 | +32.1 |

Table 2: Comparison of classification accuracies (%) between LaSeR and Llava on CIFAR-10 and CIFAR-100 datasets with 20%, 30%, and 40% mismatch ratios. The column titled "Gap" represents the gain in accuracy of LaSeR over Llava.

## B  COMPARISON WITH ZERO-SHOT FOUNDATION VLMS

An argument could be made that instead of integrating foundation models (LLMs, VLMs) into the open-set active learning pipelines, we could directly prompt multi-modal foundation models, such as Llava Liu et al. (2023), for zero-shot classification of test data. This can completely avoid all the selection and training required for open-set active learning. We evaluated Llava for zero-shot classification on the CIFAR-10 and CIFAR-100 datasets, with 20%, 30%, and 40% mismatch settings. Table 2 shows the difference in classification accuracy between the two models. In all cases, LaSeR outperforms Llava, with significantly higher margins on the CIFAR-100 dataset. These results demonstrate that while multimodal foundation models can perform relatively well on smaller datasets (CIFAR-10), their zero-shot performance starts to deteriorate on bigger, more complex datasets (CIFAR-100). These results further confirm the significance of our proposed method to effectively address OSAL.

## C  RECALL RESULTS FOR CIFAR-100

Recall is the ratio of selected samples from the relevant classes to the total number of relevant samples in a dataset. Most other methods did not report recall results, and we include only the ones that were reported in the original papers. Figures 7 compares recall of all methods on the CIFAR-100 dataset. LaSer shows much better recall than all AL methods, except LfOSA and EOAL. However, we note that the reason for a lower recall is that all other methods use the initial labeled dataset of relevant classes, while LaSeR does not. For example, for CIFAR-100, the initial dataset size is 8%, which is quite significant, as an AL model could take two AL rounds to achieve 8% recall. Despite this, LaSeR is still able to achieve comparable recall to other methods in most settings. For references, we also recalculated recall scores for other OSAL methods without considering the initial labeled data (Figure 8).These results confirm that without the initial labeled set, LaSeR outperforms all other OSAL methods in terms of recall through all AL rounds. Recall results were not reported by most SOTA methods, such as EOAL and MQ-NET on CIFAR-10 and Tiny-ImageNet, and thus are not discussed here.

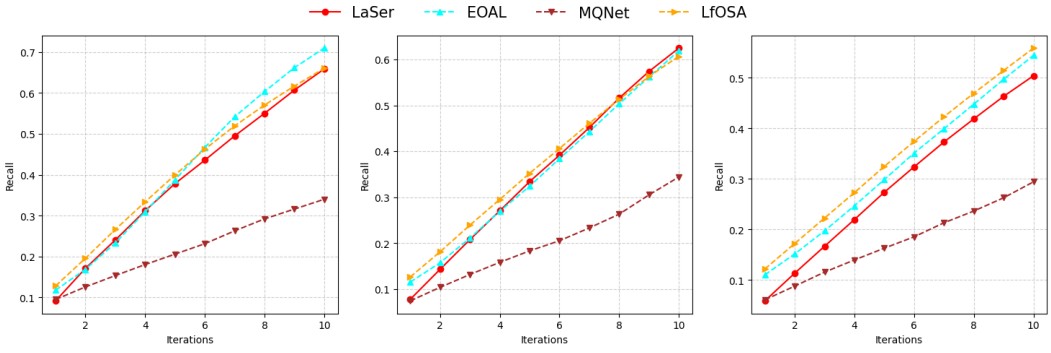

Figure 7: Recall results CIFAR-100. First, second, and third columns show recall plots for 20%, 30%, and 40% mismatch ratios

| Iteration | Recall |
|-----------|--------|
| 1 | 0.1025 |
| 2 | 0.1841 |
| 3 | 0.2437 |
| 4 | 0.3111 |
| 5 | 0.3828 |
| 6 | 0.4561 |
| 7 | 0.5126 |
| 8 | 0.5813 |
| 9 | 0.6563 |
| 10 | 0.7226 |

Table 3: Recall results for ICUB on 20% mismatch

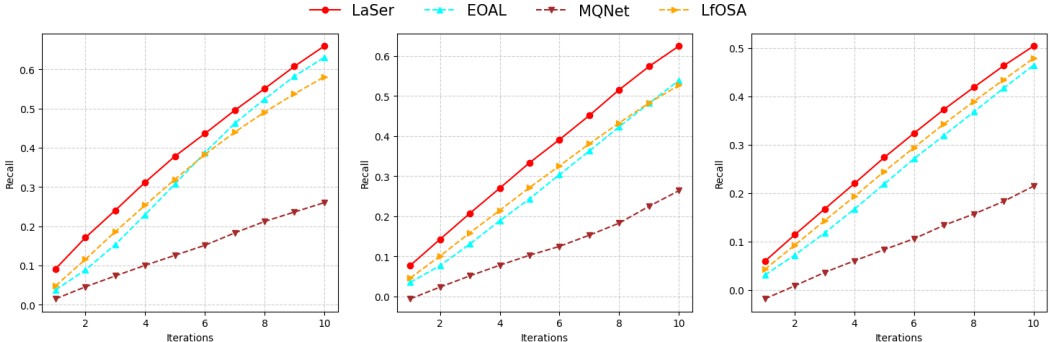

Figure 8: Recall results for CIFAR-100 when ignoring initial sampling. First, second, and third columns show recall plots for 20%, 30%, and 40% mismatch ratios

## D    RECALL RESULTS FOR ICUB

This section contains the recall results for the 20% mismatch ratio experiment done on ICUB that was describe in Section 4

Table 3 shows the recall results for ICUB dataset. The total relevant samples were 13000 out of the 65000 total samples. LaSeR managed to select 9394 samples in 10 AL rounds achieving 72.26% recall

