# OpenReview forum: "LLM-assisted Semantic Reasoning for Open-Set Active Learning"
_ICLR.cc/2026/Conference — Submitted to ICLR 2026_

### Official Review · Reviewer_S3yS · 2025-10-24

**Soundness:** 2
**Presentation:** 2
**Contribution:** 2
**Rating:** 4
**Confidence:** 3

**Summary:**

The paper proposes LaSeR, an active learning framework for open-set image classification that operates without any labeled seed data, requiring only class names as input. The framework leverages LLMs to generate textual descriptions for each known class along with additional confuser classes representing likely irrelevant concepts, then scores unlabeled images using a VLM that measures similarity between images and relevant versus confuser descriptions. After the first round, LaSeR trains a detector to distinguish relevant from irrelevant samples and combines both the VLM and detector signals through linear interpolation. Experiments on CIFAR10/100 and Tiny-ImageNet show better early-round performance.

**Strengths:**

1. The proposed method removes labeled seeds and uses only class names and LLM-generated confusers.
2. The authors address the bottleneck of label scarcity in open-set active learning by leveraging available LLMs and VLMs.

**Weaknesses:**

1. The paper uses a linear combination of the VLM and detector scores with a coefficient \delta that is manually scheduled.
But, the text and equation are inconsistent: the paper states that “we gradually decrease \delta to rely more on the detector,” but the formula indicates the opposite (larger \delta gives more detector weight).
2. The method does not explicitly enforce diversity among the queried samples. Although the authors use multiple textual prompts, the selection strategy still picks the top-n samples by score, which may lead to redundancy.
3. The evaluation scope is limited to small low-resolution datasets. The method’s scalability and robustness remain unclear on larger datasets.

**Questions:**

1. The text says that \delta is decreased each round to rely more on the detector, but according to Eq. 5, increasing \delta gives the detector more weight. Could you clarify which version is implemented, and show how performance varies with different \delta schedules?
2. Do LLM-generated confusers sometimes include classes that never appear in the unlabeled pool? If so, does this harm the relevance score? Have you tried filtering out or down-weighting confusers with consistently low similarity to any image?
3. The ablation (Table 5) shows that multiple text prompts improve performance, suggesting some diversity benefit. Did you measure redundancy among the selected samples (e.g., feature overlap)? Would adding a diversity-based selection criterion further improve label efficiency?
4. Each round trains two networks for 300 epochs. How much additional training time does this require compared to standard OSAL baselines?

---

> ### Author Response · Authors · 2025-11-26
>
> The authors would like to sincerely thank the reviewer for their helpful and constructive comments, which helped us to improve the paper. We believe that we addressed all the weaknesses/questions. We are listing our responses below and how we updated our paper based on the reviewer's suggestions.
>
> **Questions**
>
> **[Q1] Typos and different delta schedules:** There was a typo, and we fixed it in the paper (L253). The delta value is incrementally increased over the course of AL rounds, giving detector scores more weight. We have also added an experiment in Section 4.6 (Ablation Study), demonstrating the effect of increasing delta in a heuristic and more principled way (Figure 5, 6 and L518-527). We noticed a minimal change in accuracy and precision for LaSeR in all AL rounds, demonstrating that the model might not be highly sensitive to the choice of the rate of increase for the delta values.
>
> **[Q2] LLM generated confusers may not appear in the unlabeled pool:** Yes, it is possible that LLM-generated confusers can have classes that never appear in the dataset because the model does not have any knowledge about the future irrelevant classes in the environment. In the first AL round, the number of such confusers is maximal because the model does not yet know which irrelevant classes are actually present in the dataset. In later AL rounds, once the model has identified irrelevant class names through orcale labeling, we filter out and replace confusers corresponding to classes that are not present in the dataset, thereby mitigating any potential negative impact.
>
> **[Q3] Redundancy in relevant samples and diversity-based selection:** Thank you for your question. We did allow the model to select diverse samples by allowing an equal number of samples selected per textual description for the relevant classes, which have the highest relevance scores. We have updated the text in the paper to further clarify this part (L209-211). However, when the textual descriptions were removed, we noticed a high feature overlap among the selected samples, as the model selected all samples that were similar to a single textual embedding generated by the VLM.
>
> **[Q4] Additional training compared to SOTA:** We believe there might be a misunderstanding. All other OSAL methods, e.g. EOAL, train two models (a detector and a classifier) for the same number of epochs. Therefore, LaSeR does not use any additional training compared to standard OSAL baselines.
>
> **Wekanesses**
>
> **[W1, 2]** Weaknesses 1 and 2 have been addressed in the questions.
>
> **[W3] Evaluation on low resolution datasets:** Thank you for suggesting evaluating the robustness and scalability of our method on higher resolution datasets. Based on all other reviewers' suggestions, we evaluated LaSeR on a robotics dataset (iCubWorld) (Section 4.5 in the paper) with only a partial set of relevant class labels available in the first AL round. LaSeR was still able to quickly learn about informative and relevant class samples within a few initial AL rounds, demonstrating the robustness of our method in real-world settings.

---

### Official Review · Reviewer_o9FH · 2025-10-28

**Soundness:** 3
**Presentation:** 3
**Contribution:** 3
**Rating:** 6
**Confidence:** 5

**Summary:**

This paper introduces LaSeR (LLM-assisted Semantic Reasoning), a novel framework for open-set active learning (OSAL) that addresses the cold-start problem without requiring any initially labeled data. Unlike prior OSAL methods, LaSeR only uses textual labels of relevant classes and leverages large language models (LLMs) to generate diverse textual descriptions and semantically similar but irrelevant classes. These descriptions are then encoded via a vision-language model (VLM) to compute relevance scores for unlabeled images. In later active learning rounds, a CNN-based detector is trained on previously labeled data to further refine sample selection. The method dynamically balances VLM-based semantic reasoning and detector-based discrimination. Extensive experiments on benchmark datasets show that LaSeR outperforms state-of-the-art methods, especially in early rounds.

**Strengths:**

1. The paper introduces, for the first time, an OSAL method that operates without initial labeled data, thereby tackling the cold-start issue and bridging a gap in current research.
2. The method combines LLMs, VLMs, and CNN detectors, dynamically adapting its strategy at different stages to make full use of semantic reasoning and visual features.
3. The paper is clearly written and easy to follow. The proposed approach is simple yet effective, enhancing the interpretability and transparency of model selection by generating class descriptions and confusing classes.

**Weaknesses:**

1. The paper contains multiple language and expression errors, such as “where we DL is unavailable”, “We gradually decrease δ in later AL rounds”, and inconsistencies like “CIFAR10” vs. “CIFAR-10”, among others.
2. The paper lacks sufficient rigor, as the experimental results on CIFAR-100 and Tiny-ImageNet do not present all the compared methods comprehensively.
3. The introduction of LLMs or VLMs inherently brings substantial prior knowledge. It is therefore necessary for the authors to conduct additional validation on some non-standard datasets to better demonstrate the generalizability of the proposed method.
4. The training and score design of the detector are overly simple and are likely to underperform compared with existing methods such as EOAL or the more recent EAOA [1]. The advantage of the proposed approach mainly appears in the early stages, which also explains why its performance on CIFAR-10 becomes slightly inferior to the compared methods in later rounds.

[1] Rethinking Epistemic and Aleatoric Uncertainty for Active Open-Set Annotation: An Energy-Based Approach

**Questions:**

Please see the Weaknesses.

---

> ### Author Response · Authors · 2025-11-26
>
> The authors would like to sincerely thank the reviewer for their helpful and constructive comments, which helped us to improve the paper. We believe that we addressed all the weaknesses/questions. We are listing our responses below and how we updated our paper based on the reviewer's suggestions.
>
> **Weaknesses**
>
> **[W1] Typos and Language Errors:** We have fixed the suggested typos and language errors in the paper.
>
> **[W2] Comparison with methods on CIFAR-100 and TinyImageNet:** We added results for all the methods evaluated on CIFAR-100 and TinyImageNet from the recent SOTA paper (EOAL) for a fair comparison. We also note that we noticed a discrepancy among different methods in terms of which OSAL mismatch setups and datasets they report results on. For example, EOAL reported results on 20%, 30% and 40% mismatch ratios for all three datasets. However, BUAL reported results on 40% and 60% mismatch on the three datasets in the original paper. However, the suggested paper EAOA from CVPR 2025 included results for most methods on most mismatch cases on the three datasets, and we have included all of those results in the paper (Figure 3). We verified that the settings in EAOA are similar to EOAL, and LaSeR's performance is similar for both settings.
>
> **[W3] Experiment on non-standard dataset and real-world applicability:** Thank you for your helpful comment and for suggesting evaluation on non-standard datasets with real-world applicability. The major reason for evaluating the proposed model on CIFAR-10, CIFAR-100 and Tiny ImageNet datasets was to have a fair comparison with prior OSAL methods, which have all been evaluated on these benchmarks. However, we agree with the reviewer that the proposed method should be tested on more real-world datasets and settings, given its unique ability to work with no initial data. Therefore, we have performed an experiment on a robotics dataset (iCubWorld) (Section 4.5 in the paper) with only a partial set of relevant class labels available in the first AL round. LaSeR was still able to quickly learn about informative and relevant class samples within a few initial AL rounds, demonstrating the robustness of our method in real-world settings. The results also demonstrate that the LLM-based semantic reasoning can allow the model to find missing relevant classes as well, given a task description for a robot.
>
> **[W4] Simple Detector and Underperformance compared to SOTA, such as EOAL and EAOA:** We thank the reviewer for suggesting a more recent OSAL method from CVPR 2025. We have included comparisons with EAOA in our paper (Figure 3). We verified that the settings in EAOA are similar to EOAL, and LaSeR's performance is similar for both settings. We note that on a simpler dataset (CIFAR-10) most methods, including LaSeR, saturate to a similar final accuracy. However, for more complex datasets, CIFAR-100 and Tiny ImageNet, LaSeR consistently outperforms both EOAL and EAOA in the final AL rounds, on all mismatch setups, except the 20% mismatch setup on CIFAR-100. For the most complex dataset, Tiny ImageNet, on 20% mismatch, LaSeR outperforms EAOA by a margin of ~5% in terms of accuracy in the final AL round. Similarly, LaSeR outperforms EOAL by a margin of 10% and 3% on the 30% and 40% setups on Tiny ImageNet (EAOA did not report results for 30% and 40% mismatches on Tiny ImageNet). These results demonstrate that our method performs better than SOTA on more complex datasets (with more classes, and higher resolution images).
>
> We do agree with the reviewer that the CNN detector for our method is much simpler than other methods. However, this further demonstrates that our method, even with a simple detector, can outperform SOTA methods by using the proposed LLM-based semantic reasoning, which leads to the selection of relevant and diverse samples even in initial AL rounds. We also note that the use of a simple detector allows our model to have lower latency in the training phase than SOTA methods (EOAL), even when using LLMs and VLMs (~4 minutes and 10s for LaSeR in the first AL round on CIFAR-100 with 20% mismatch, and ~5 minutes for EOAL).

---

### Official Review · Reviewer_5HYR · 2025-10-30

**Soundness:** 3
**Presentation:** 3
**Contribution:** 4
**Rating:** 4
**Confidence:** 4

**Summary:**

This paper introduces LLM-assisted Semantic Reasoning (LaSeR), an active learning framework that addresses the challenge of being able to distinguish between, and effectively select the most relevant samples in a dataset that contains relevant and irrelevant classes. LaSeR integrates LLMs (to generated descriptions from the given labels of relevant classes) and VLMs (to measure the relevance of each image w.r.t. the label descriptions) to compute a relevance score which guides sample selection for the next active learning round. This process allows for the framework to mitigate the cold start problem, as it does not require an initial set of labeled data to guide the initial rounds of active learning. In later rounds, this framework also integrates a CNN-based detector to more effectively select the relevant and helpful samples. The authors also present experiments on popular AL datasets (such as CIFAR-10, CIFAR-100, Tiny-ImageNet) which showcase LaSeR outperforming other state-of-the-art methods in metrics like accuracy and precision, especially shinning in the early training rounds.

**Strengths:**

- The paper proposes a novel solution that addresses the cold start problem in open-set active learning which eliminates the need for initial labeled data, this ability to effectively start the AL process (and select relevant examples in early rounds) without initial labels make it especially useful in cases where labeled data is scarce and hard to find
- In the empirical experiments presented, LaSeR was showcased to outperform popular regular AL and OSAL-specific methods on a number of benchmarks (CIFAR-10, CIFAR-100, Tiny-ImageNet) in classification accuracy, especially in the early rounds (great for targeting the cold-start issue)
- Rigorous ablation studies were conducted to demonstrate the contribution and importance of each component within the proposed framework

**Weaknesses:**

- Limitations of the real-world applicability of the framework: The proposed framework seems to only be effective for image classification use cases (the framework setup rather specifically to that task, and the experiments are all conducted on image classification datasets), this limits its generalizability and utility for broader applications. The first step of the framework (using the LLM to generate descriptions / alternative class names) also assumes informative class names that LLMs can easily understand and operate on, which might further limit its effectiveness under situations where the class names are less informative / meaningful. Lastly, the evaluations were conducted on small scale image datasets (two of which, CIFAR-10 / CIFAR-100, are also closely related), which might not give a full representation of real-world settings.
- The proposed framework is heavily dependent on LLMs for generating class descriptions and class labels, however the computational costs and latency of these LLM calls are not reported or analyzed. It is unclear what the tradeoffs are and how they compare to the baseline methods, which is important for real-world use cases especially for applications with limited budget.
- There is minimal theoretical grounding to the formulation of some components of the proposed framework. Equations 1 and 4 are not rigorously derived (for example, why use subtraction between the components, instead of other potential measures such as taking the ratio / softmax / etc). The final relevance score is also a weighted average of the VLM-based and CNN-based score with an increasing weight on the CNN-based detector scores (linearly by 0.1 each round), however this weighting strategy also does not have a clear theoretical backing and it is unclear if the proposed rate of increase is optimal.

**Questions:**

- As mentioned above, there is a lack of discussion of the extra costs and latency incurred by the LLM / VLM calls. Can you provide some estimates and analysis for the extra computational costs + latency for these calls (and how they might scale with larger datasets with large number of classes)? It would also be helpful to provide some comparisons about the additional costs here vs cost savings from the reduced number of annotations required to achieve a targeted level of performance.
- The detector model is trained over n+1 classes, I am curious if treating all additional classes as a single “irrelevant” class is the best strategy here? I can see how lumping classes that might be pretty difference from one another can lead to poor decision boundaries. Has the authors experiments with alternative setups (eg. treating them as separate classes or group of classes)? I also wonder if the number of “irrelevant” classes increase, would this cause a massive imbalance in the label distribution?
- The selection precision for LaSeR is often outperformed by methods such as LfOSA or EOAL (especially beyond the first few AL rounds, and the accuracy performance of these baseline methods also sometimes become comparable with LaSeR in the later rounds). It seems like as we continue into the later AL rounds, LaSeR becomes less effective, do the authors think there might be some early stopping mechanism, or have suggestions on how to identify the point at which LaSeR’s performance gains are maximized before diminishing?
- [not a question but pointing out an error]
    > “We gradually decrease δ in later AL rounds shifting the balance of the selection process toward the detector.”

  In page 5, there is a typo in the paragraph below eq. 5, “decrease” here should be “increase”

---

> ### Author Response · Authors · 2025-11-26
>
> The authors would like to sincerely thank the reviewer for their helpful and constructive comments, which helped us to improve the paper. We believe that we addressed all the weaknesses/questions. We are listing our responses below and how we updated our paper based on the reviewer's suggestions.
>
> **Questions**
>
> **[Q1] Extra costs incurred by the LLM/VLM calls:** Thank you for your question. We wanted to clarify that only a single LLM call is made in the training stage of the first AL round. Additionally, another LLM call is only made once in the training stage of later AL rounds if a new irrelevant class is encountered and more textual descriptions are generated for that irrelevant class. This leads to an extremely small addition (~5s per LLM call) of latency during the training stage with the GPT-4o mini. Smaller LLM models could further reduce this latency. Additionally, for comparison with other non-LLM based methods, such as EOAL (previous SOTA), LaSeR takes ~10s for two LLM calls in the first AL round, ~2 minutes to get image and sentence embeddings from CLIP for entire dataset, and ~2 minutes for 300 epochs of training on the annotated data on the 20% mismatch setup on the CIFAR-100 dataset (total 4 minutes and 10 seconds of time in the training phase of the first AL rounds). EOAL, on the other hand, took ~5 minutes in the first AL round on the same dataset, and 300 epochs of training. These results confirm that our model still takes less time for training than other SOTA methods. Finally, there is no latency during the test phase, because only the CNN classifier is used for this phase without any LLM or VLM calls.
>
> **[Q2] Combining Irrelevant classes into a single class for the detector:** Thank you for this suggestion and your question. Since our method is quite effective in selecting relevant classes even in the first AL round (high selection precision), only a small number of irrelevant class images are selected. For example, for the first AL round on 40% mismatch setup on the CIFAR-10 dataset, LaSeR's selection precision is 90%. In this round, for a query batch size of 1500 samples, this leads to selection of 1350 relevant class samples, resulting in ~337 samples per relevant class and ~25 samples for each irrelevant class. If we use the irrelevant classes separately, there would be a huge class imbalance, but merging all of them together in a single irrelevant class reduces this imbalance (150 total irrelevant samples).
>
> **[Q3] Performance of LaSeR in later AL rounds:** OSAL methods are designed for real-world cases when data and the number of irrelevant classes might be unlimited, however, in the benchmark datasets used for OSAL evaluations (e.g. CIFAR-10) there are only a limited number of classes available, which results in saturation of results in later AL rounds for most OSAL (and traditional AL) methods. The major limiting factor for most OSAL methods is initial AL rounds when data is limited (or in our case, no data is available). Our method is mainly designed to tackle the OSAL problem in earlier rounds, with the use of generative and semantic reasoning capabilities of LLMs. In the later AL rounds, most of the performance is dependent on the CNN-based detector, which usually results in similar (or even lower) performance than other methods that use more complex detectors. Thank you for suggesting identifying the point when LaSeR’s performance gains are maximized. We could track the precision for both the detector and LLM-VLM parts separately in continuous AL rounds, and determine when there is a high saturation. However, this would be dependent on the dataset and the environment, and how saturated the environment might be.
>
> **[Q4] Typos:** We have fixed this typo in the paper (L253).
>
> **Weaknesses**
>
> **[W1] Limitations of real-world applicability of the framework:** Thank you for your helpful comment and for suggesting evaluation on non-standard datasets with real-world applicability. The major reason for evaluating the proposed model on CIFAR-10, CIFAR-100 and Tiny ImageNet datasets was to have a fair comparison with prior OSAL methods, which have all been evaluated on these benchmarks. However, we agree with the reviewer that the proposed method should be tested on more real-world datasets and settings, given its unique ability to work with no initial data. We also agree with the reviewer that full information about relevant class labels in the initial AL round might also be challenging in real-world settings, particularly household robotics applications. Therefore, we have performed an experiment on a robotics dataset (iCubWorld) (Section 4.5 in the paper) with only a partial set of relevant class labels available in the first AL round. LaSeR was still able to quickly learn about informative and relevant class samples within a few initial AL rounds, demonstrating the robustness of our method in real-world settings.
>
> **[W2]:** Weakness 2 has been addressed in the questions.

---

> ### Author Response · Authors · 2025-11-26
>
> **[W3] Theoretical grounding of equations:** We agree with the reviewer that equations 1 and 4 were designed heuristically and were not based on a theoretical foundation. We agree that taking a ratio instead of subtraction could be a viable option. We tried the ratio-based method, but it failed to produce meaningful results due to a significant variance in similarity values and division by small similarity values. We will add these results to the appendix of the final version of the paper. Additionally, we agree that an incremental increase of the values of the delta parameter in equation 5 is also heuristic. We have, however, developed a way to determine the delta value directly from the relevance scores generated by the LLM-VLM stage and the detector. We can take the ratio of the detector precision and the sum of the detector and VLM precisions to determine a principled way to find the delta value. In this way, as the LLM-VLM scores become less effective compared to the detector in later AL rounds, the delta value is shifted in the detector’s direction. Results in Section 4.6 (Ablation study, L518-527) and Figures 5 and 6 show the effect of using the precision-based delta scores. We noticed a minimal change in accuracy and precision for LaSeR in all AL rounds, demonstrating that the model might not be highly sensitive to the choice of the rate of increase for the delta values.

---

### Official Review · Reviewer_pcof · 2025-10-31

**Soundness:** 3
**Presentation:** 3
**Contribution:** 2
**Rating:** 2
**Confidence:** 4

**Summary:**

This paper introduces LaSeR, a novel method for Open-Set Active Learning that distinctively tackles the "cold-start" problem by not requiring any initial labeled data. The core idea is to leverage the semantic reasoning of Large Language Models to generate textual descriptions for both relevant classes and, crucially, potentially confusing irrelevant classes. A Vision-Language Model is then used to compute a relevance score for unlabeled images based on these descriptions . In later rounds, the method adaptively integrates this score with a standard CNN-based detector trained on the newly acquired labels.

**Strengths:**

1. The paper addresses two key, practical challenges in active learning: the presence of irrelevant data (open-set) and the cold-start problem (no initial labels). Tackling OSAL without any initial labeled set is a valuable and challenging research direction.

2. The core idea of using an LLM to proactively generate descriptions of confusing irrelevant classes (e.g., "bird" for the relevant class "airplane") is a clever way to improve the VLM's zero-shot filtering capability.

3. The experimental results demonstrate that LaSeR outperforms baselines in the initial AL rounds, validating its effectiveness at solving the cold-start problem where other methods traditionally struggle.

**Weaknesses:**

1. The main criticism is that the overall construction is a fairly direct engineering integration of existing pre-trained models, the method's effectiveness hinges almost entirely on the powerful, pre-trained LLM and VLM, which, however, is expected to be better than the compared methods, since more information has been used. The contribution feels incremental.

2. The claim of no initial labeled data is qualified by requiring full knowledge of relevant class names; many OSAL setups do not assume this, so the setting is easier in some respects than standard OSAL.

3. In the experiments, the paper says "We report results of all OSAL methods from their original papers, if available," suggesting some curves are not re-run under this paper's exact setup (splits, temperature, budgets), which complicates fairness. Besides, the paper cites recent OSAL works like Mao et al. (2024) and Zong et al. (2024) in the introduction , but these are conspicuously absent from the experimental baselines. Finally, a simple CLIP-only acquisition rule using only class names (no LLM aug) would be a very relevant ablation/baseline.

**Questions:**

Please see Weaknesses

---

> ### Author Response · Authors · 2025-11-26
>
> The authors would like to sincerely thank the reviewer for their helpful and constructive comments, which helped us to improve the paper. We believe that we addressed all the weaknesses/questions. We are listing our responses below and how we updated our paper based on the reviewer's suggestions.
>
> **Weaknesses**
>
> **[W1] Use of pre-trained LLM and VLM:** We agree that our proposed model relies on LLM reasoning and VLM embeddings; however, unlike other methods, we do not use any prior knowledge in terms of a full labelled dataset of relevant classes to deal with the cold-start problem. Additionally, the use of LLM-based reasoning allows our model to be applicable on non-standard and more unconstrained real-world settings, such as robotics applications, where the robot might be exploring its environment to find relevant and informative data. We have an additional experiment on a robotics dataset (iCubWorld) in Section 4.5 of the paper, demonstrating the effectiveness of our method even when a full set of relevant class labels is unavailable. Additionally, we demonstrated in Appendix B that using foundation models directly for zero-shot classification leads to significantly inferior results compared to our proposed method, indicating that using powerful pre-trained VLMs or LLMs does not directly solve the problem. Finally, we note that the LLM is prompted only once during the first AL round and prompted in later rounds only if there are any new irrelevant classes observed, and new descriptions need to be generated for those classes, indicating limited reliance on the LLM.
>
> **[W2] Relevant class names in OSAL setups:** We believe there is a misunderstanding, all OSAL setups (e.g. (Park et al. 2022), (Safaei at el. 2024)) not only assume full knowledge of relevant class names, but also assume the availability of a small, fully annotated dataset for all relevant classes before the first AL round. However, this comment helped us further analyze our method under stricter constraints when only a partial set of relevant classes is known, making it an even harder OSAL setting. We conducted an experiment on the iCubWorld dataset (Section 4.5 in the paper) with only a partial set of relevant class labels available in the beginning. LaSeR was still able to quickly learn about informative and relevant class samples within a few initial AL rounds, demonstrating the robustness of our method in real-world settings. We thank the reviewer for their helpful comment suggesting that we further demonstrate the utility of our method in more challenging OSAL settings, where prior methods may be less suitable.
>
> **[W3] Fair comparison with other methods:** We believe there might be a misunderstanding because of the way we described this in our paper. We followed the same dataset, implementation, and experimental settings as used in EOAL (Safaei et al. 2024) (recent state of the art), which allowed for a fair comparison against EOAL and other OSAL methods reported in that paper. Additionally, we have reported results by a more recent paper (EAOA) suggested by reviewer o9FH, as well as results for other methods reported in that paper, and our proposed method (LaSeR) outperforms all other methods in most datasets and mismatch settings (Figure 3). We verified that the settings in EAOA are similar to EOAL, and LaSeR's performance is similar for both settings. We also note that Mao et al. (2024) and Zong et al. (2024) did not report results on most setups considered in our paper and in the EOAL paper, which is why they were not included in Figure 3. However, they have now been added based on the experimental results from the EAOA paper (Zong and Huang 2025).
>
> **[W3] CLIP-Only baseline:** Thank you for suggesting evaluating using a simple CLIP-only baseline without using LLM-augmentations. We have added two versions of our method in Section 4.6 (Ablation Studies) where the LLM is removed but the detector and VLM are kept, and another version where the detector is also removed and only the VLM is kept. The results (Figure 5) show ~10% decline in accuracy when LLM-generated augmentations are removed.

---

### Author Response · Authors · 2025-12-03
**Summary of the Rebuttal and General Response**

We would like to express our sincere gratitude to the AC and all reviewers for the time and effort they have invested during the review and rebuttal phases. Given the limited time before the discussion period concluded, we provide a concise summary of our manuscript, main strengths of our work noted by the reviewers, and major and minor categories of reviewer comments/questions and how we addressed them in the paper. We remain fully available for any additional inquiries by the AC.

**Summary:** Our paper introduces LaSeR, a novel method for Open-Set Active Learning that distinctively tackles the "cold-start" problem by not requiring any initial labeled data. The core idea is to leverage the semantic reasoning of Large Language Models to generate textual descriptions for both relevant classes and, crucially, potentially confusing irrelevant classes. A Vision-Language Model is then used to compute a relevance score for unlabeled images based on these descriptions. In later rounds, the method adaptively integrates this score with a standard CNN-based detector trained on the newly acquired labeled data. Extensive experiments on popular OSAL datasets (such as CIFAR-10, CIFAR-100, Tiny-ImageNet) demonstrate LaSeR outperforming other state-of-the-art methods in metrics like accuracy and precision.

**Strengths noted by reviewers:**
- All reviewers highlighted that our paper presents **a novel method to address two main practical challenges in open-set active learning (OSAL)**: the presence of irrelevant data (open-set) and the cold-start problem (no initial labeled data). Particularly, our paper is the first to address the OSAL problem without any initial labeled data, thus bridging a research gap in the field (Reviewer o9FH), and it is especially useful in cases where labeled data is scarce and hard to find (Reviewer 5HYR).
- Reviewers pcof, o9FH, and S3yS highlighted the core idea of our paper, **integrating semantic reasoning and generative ability of LLMs with VLMs, and CNN detectors**, dynamically adapting the sample selection strategy at different stages to address the bottleneck of label scarcity in OSAL.
- **LaSeR was showcased to outperform popular regular AL and OSAL-specific methods** on a number of benchmarks (CIFAR-10, CIFAR-100, Tiny-ImageNet) in classification accuracy, especially in the early rounds (great for targeting the cold-start issue). (Reviewers pcof and o9FH).
- **Rigorous ablation studies** were conducted to demonstrate the contribution and importance of each component within the proposed framework. (Reviewer 5HYR)
- The paper is **clearly written and easy to follow**. (Reviewer o9FH)
- The proposed **approach is simple yet effective, enhancing the interpretability and transparency** of model selection by generating class descriptions and confusing classes. (Reviewers pcof, o9FH, S3yS)

**Major Reviewer Comments/Questions**
Following are major comments/questions by most reviewers, followed by our responses and how we updated the paper based on reviewer suggestions.
- **Real-World Applicability and Evaluation on Non-Standard Benchmarks** *All reviewers are in agreement that given the unique nature of our method to not rely on any initial labeled data and reliance on LLMs and VLMs, it must also be evaluated on non-standard datasets, to better demonstrate its real-world applicability. Reviewers also raised the concern that in the real-world, even a full set of relevant class labels might not be available (our method relies on relevant class names in the beginning). Reviewer pcof also raised a concern that other OSAL methods and setups do not assume full knowledge of relevant class labels.*
We thank the reviewers for suggesting evaluation of LaSeR on non-standard datasets with real-world applicability. The main reason for evaluating the proposed model on CIFAR-10, CIFAR-100 and Tiny ImageNet datasets was to have a fair comparison with prior OSAL methods, which have all been evaluated on these benchmarks. We agree with reviewer suggestions, and, therefore, have performed an experiment on a robotics dataset (iCubWorld) (Section 4.5 in the paper) with only a partial set of task-relevant class labels available in the first AL round. LaSeR was still able to quickly learn about informative and relevant class samples within a few initial AL rounds (achieving 93.6% accuracy within the first 4 AL rounds), demonstrating the robustness of our method in real-world settings.
Additionally, regarding reviewer pcof’s comment, we wanted to clarify that all other OSAL methods not only use full knowledge of relevant class labels, but also but also assume the availability of a small, fully annotated dataset for all relevant classes before the first AL round.

---

> ### Author Response · Authors · 2025-12-03
> **Major Reviewer Comments/Questions - Continued**
>
> - **Fair Comparison with SOTA OSAL Methods** *Reviewers asked if the experimental setting in our paper was similar to the ones used by previous works. They also asked why some methods were missing in Figure 3 for experiments on CIFAR-10, CIFAR-100, and Tiny ImageNet datasets.* We believe there might be a misunderstanding. We followed the same dataset, implementation, and experimental settings as used in EOAL (Safaei et al. 2024) (recent SOTA), which allowed for a fair comparison against EOAL and other OSAL methods reported in that paper. We observed that different methods report results on different subsets of mismatch ratios and datasets. For example, EOAL reported results on 20%, 30% and 40% mismatch ratios for all three datasets, while BUAL reported results on 40% and 60% mismatch in its original paper, which is why some of the methods were omitted in the settings where no results were available in the original papers. However, the suggested paper by reviewer o9FH, EAOA from CVPR 2025, included results for most methods on most mismatch cases on the three datasets, and we have included all of those results in the paper (Figure 3). We verified that the settings in EAOA are similar to EOAL, and LaSeR's performance is similar for both settings. Regarding performance comparison, we note that on a simpler dataset (CIFAR-10) most methods, including LaSeR, saturate to a similar final accuracy. On the more challenging CIFAR-100 and Tiny ImageNet datasets, LaSeR consistently outperforms both EOAL and EAOA in the final AL rounds for all mismatch setups, except the 20% mismatch on CIFAR-100. Particularly, for the most complex dataset, Tiny ImageNet, on 20% mismatch, LaSeR outperforms SOTA methods by margins of ~5%, 10% and 3% on the 20%, 30% and 40% setups, respectively. These results demonstrate that our method performs better than SOTA on datasets with more classes and higher resolution images.
> - **Additional Training and Extra Costs Incurred by the VLM/LLM Calls** *Reviewer5HYR asked if there are additional computational costs associated with LLM/VLM calls, and if that causes latency and more training time compared to SOTA methods? Reviewer S3yS also asked if there is additional training time required for LaSeR compared to OSAL baselines.*
> We wanted to clarify that only a single LLM call is made in the first AL round of training, and another LLM call is only made once in later AL rounds if a new irrelevant class is encountered and more textual descriptions are generated for that irrelevant class. This leads to an extremely small addition (~5s per LLM call) of latency during the training stage with the GPT-4o mini. Additionally, for comparison with other non-LLM based methods, such as EOAL (previous SOTA), LaSeR takes ~10s for two LLM calls in the first AL round, ~2 minutes to get image and sentence embeddings from CLIP for entire dataset, and ~2 minutes for 300 epochs of training on the annotated data on the 20% mismatch setup on the CIFAR-100 dataset (total 4 minutes and 10 seconds). EOAL, on the other hand, took ~5 minutes in the first AL round on the same dataset, and 300 epochs of training. These results confirm that LaSeR still takes **less time** for training than other SOTA methods. Regarding any additional training, all OSAL methods, e.g. EOAL, including LaSeR, train two models (a detector and a classifier) for the same number of epochs. Therefore, LaSeR does not use any additional training compared to standard OSAL baselines.
> - **Theoretical grounding and sensitivity to the rate of increase of delta values** *Reviewer5HYR raised a concern that there is minimal theoretical grounding to the formulation of some equations for our method. Particularly, the weighting strategy (delta value in equation 5) to balance CNN detector and LLM-VLM scores is heuristic and might not be optimal. Reviewer S3yS also asked how performance varies with different delta value schedules.* We have developed a principled way to determine the delta value directly from the relevance scores generated by the LLM-VLM stage and the detector. We can take the ratio of the detector precision and the sum of the detector and VLM precisions to determine a principled way to find the delta value. In this way, as the LLM-VLM scores become less effective compared to the detector in later AL rounds, the delta value is shifted in the detector’s direction (L518-527 in the paper). Results in Section 4.6 (Ablation study) and Figures 5 and 6 show the effect of using the precision-based delta scores. We noticed a minimal change in accuracy and precision for LaSeR in all AL rounds, demonstrating that the model might not be highly sensitive to the choice of the rate of increase for the delta values.
> - **Typos** *All reviewers pointed out a few typos in the paper, particularly the change in delta values in equation 5.*
> We have fixed all typos in the paper, particularly we fixed the language regarding delta values in equation 5 (L253 in the paper).

---

> > ### Author Response · Authors · 2025-12-03
> > **Reviewer Specific Questions/Comments**
> >
> > Following are comments/questions by different reviewers, followed by our responses and how we updated the paper based on reviewer suggestions.
> >
> > - **Use of Pre-trained VLM and LLM:** *Reviewer pcof raised the concern that our method relies on pre-trained VLMs and LLMs, and is expected to be better than compared methods. Similar concern was also shared by reviewer o9FH and they suggested evaluating LaSeR on a non-standard dataset.* We explain experiments on a non-standard dataset in response to major comments by the reviewers above.
> > We further wanted to state that while our model does rely on pre-trained LLMs and VLMs, it does not rely any prior knowledge in terms of a full labelled datasets of relevant classes. Additionally, we demonstrated in Appendix B that using pre-trained foundation models directly for zero-shot classification leads to significantly inferior results compared to our proposed method as well as some other OSAL baselines, indicating that using pre-trained VLMs or LLMs does not directly solve the problem.
> > - **CLIP-Only Baseline:** *Reviewer pcof also asked to include a simple CLIP-only baseline in ablation studies.* We have included two versions of LaSeR with the CLIP model and without the LLM in Ablation Studies (Section 4.6). The results (Figure 5) show ~10% decline in accuracy when LLM-generated augmentations are removed.
> > - **Combining Irrelevant classes into a single class for the detector:** *Reviewer 5HYR asked if combining all irrelevant classes into a single class for the CNN detector would cause an imbalance.* Since our method is quite effective in selecting relevant classes even in the first AL round (high selection precision), only a small number of irrelevant class images are selected. For example, for the first AL round on 40% mismatch setup on the CIFAR-10 dataset, LaSeR's selection precision is 90%. In this round, for a query batch size of 1500 samples, this leads to selection of 1350 relevant class samples, resulting in ~337 samples per relevant class and ~25 samples for each irrelevant class. If we use the irrelevant classes separately, there would be a huge class imbalance, but merging all of them together in a single irrelevant class reduces this imbalance (150 total irrelevant samples).
> > - **LLM-generated confusers may not appear in the unlabeled pool:** *Reviewer S3yS asked if LLM-generated confusers might include classes that never appear in the unlabeled pool, and if this harms the relevance score and if we tried filtering out or down-weighting confusers with consistently low similarity to any image.*
> > It is possible that LLM-generated confusers can have classes that never appear in the dataset because the model does not have any knowledge about the future irrelevant classes in the environment. In the first AL round, the number of such confusers is maximal because the model does not yet know which irrelevant classes are actually present in the dataset. In later AL rounds, once the model has identified irrelevant class names through oracle labeling, we filter out and replace confusers corresponding to classes that are not present in the dataset, thereby mitigating any potential negative impact.
> > - **Redundancy in relevant samples and diversity-based selection:** *Reviewer S3yS asked if we measured redundancy among the selected samples and if a diversity-basd selection criterion further improve label efficiency.*
> > We did allow the model to select diverse samples by allowing an equal number of samples selected per textual description for the relevant classes, which have the highest relevance scores. We have updated the text in the paper to further clarify this part (L209-211). However, when the textual descriptions were removed, we noticed a high feature overlap among the selected samples, as the model selected all samples that were similar to a single textual embedding generated by the VLM.

---

### Meta-Review · Area_Chair_HDzG · 2026-01-07

**Summary:**

This paper proposes LaSeR, a novel approach to Open-Set Active Learning that explicitly addresses the cold-start problem by eliminating the need for any initially labeled data. The central idea is to exploit the semantic reasoning capabilities of large language models to generate textual descriptions for both relevant classes and, importantly, potentially confusing irrelevant classes. A vision–language model is then employed to compute relevance scores for unlabeled images based on these descriptions. The authors evaluate LaSeR on widely used active learning benchmarks, including CIFAR-10, CIFAR-100, and Tiny-ImageNet. The experimental results demonstrate that LaSeR consistently outperforms state-of-the-art baselines in terms of accuracy and precision, with particularly strong performance in the early training rounds.

The paper received reviews from four reviewers, and the majority judged that it does not meet ICLR’s acceptance standards.

**Reviewer Concerns:**

During the rebuttal phase, the authors addressed the following concerns:
- The originality of the proposed method;
- The appropriateness of the experimental setup;
- The reliability of the experimental results.

**Reviewer Scores:**

Although the authors responded to the reviewers’ comments, certain issues related to the methodology and experiments remain, and the reviewers did not revise their scores.

---

> ### Public Comment · ~Ali_Ayub1 · 2026-05-04
> **AC didn't even know the updated ICLR policy**
>
> The AC said, "...the reviewers did not revise their scores..." How are the reviewers supposed to revise their scores when they are blocked from responding or updating anything? How is the AC not aware of this policy?

---

### Decision · Program_Chairs · 2026-01-26

Reject